# GNSS-Free Outdoor Localization Techniques for Resource-Constrained IoT Architectures: A Literature Review

Azin Moradbeikie [1,2], Ahmad Keshavarz [1], Habib Rostami [1], Sara Paiva [2] and Sérgio Ivan Lopes [2,3,*]

1  IoT and Signal Processing Research Group, ICT Research Institute, Faculty of Intelligent Systems Engineering and Data Science, Persian Gulf University, Bushehr 7516913817, Iran; amoradbeiki@pgu.ac.ir (A.M.); a.keshavarz@pgu.ac.ir (A.K.); habib@pgu.ac.ir (H.R.)
2  ADiT-Lab, Instituto Politécnico de Viana do Castelo, Rua Escola Industrial e Comercial Nun' Alvares, 4900-347 Viana do Castelo, Portugal; sara.paiva@estg.ipvc.pt
3  IT — Instituto de Telecomunicações, Campus Universitário de Santiago, 3810-193 Aveiro, Portugal
*  Correspondence: sil@estg.ipvc.pt; Tel.: +351-258-819-700

**Abstract:** Large-scale deployments of the Internet of Things (IoT) are adopted for performance improvement and cost reduction in several application domains. The four main IoT application domains covered throughout this article are smart cities, smart transportation, smart healthcare, and smart manufacturing. To increase IoT applicability, data generated by the IoT devices need to be time-stamped and spatially contextualized. LPWANs have become an attractive solution for outdoor localization and received significant attention from the research community due to low-power, low-cost, and long-range communication. In addition, its signals can be used for communication and localization simultaneously. There are different proposed localization methods to obtain the IoT relative location. Each category of these proposed methods has pros and cons that make them useful for specific IoT systems. Nevertheless, there are some limitations in proposed localization methods that need to be eliminated to meet the IoT ecosystem needs completely. This has motivated this work and provided the following contributions: (1) definition of the main requirements and limitations of outdoor localization techniques for the IoT ecosystem, (2) description of the most relevant GNSS-free outdoor localization methods with a focus on LPWAN technologies, (3) survey the most relevant methods used within the IoT ecosystem for improving GNSS-free localization accuracy, and (4) discussion covering the open challenges and future directions within the field. Some of the important open issues that have different requirements in different IoT systems include energy consumption, security and privacy, accuracy, and scalability. This paper provides an overview of research works that have been published between 2018 to July 2021 and made available through the Google Scholar database.

**Keywords:** IoT; IIoT; LPWAN; LoRaWAN; GNSS-free; outdoor localization; smart cities; smart transportation; smart healthcare; smart manufacturing



## 1. Introduction

Internet of Things (IoT) and Industrial Internet of Things (IIoT) applications typically consume data generated by devices that have a particular spatio-temporal context. To increase its applicability, data generated by the IoT devices need to be time-stamped (time context) and in some application cases, it also needs to be spatially contextualized. If on the one hand, the addition of time context to data (generating time-stamped series data) has no relevant additional cost, on the other hand, the addition of location context to the time-series data has some known associated costs that may be related to: (1) operational cost (infrastructure-dependent); (2) autonomy (energy consumption); (3) size (equipment, batteries, antennas); (4) price (tag/communications technology); and (5) accuracy level [1]. Moreover, location data are of extreme importance in business intelligence, with direct applications in traceability, logistics, manufacturing, Industry 4.0, among others [2–4].

Moreover, IoT technologies provide a promising opportunity to build powerful and effective industrial systems and applications [5]. IoT is built by adding communication and computation ability to devices. Many industries owners do not agree with the entrance of IoT to the industries but with the addition of communication and computation ability to different devices, we must know that IoT is already a part of our industry and life. IoT has impacts on almost all aspects of our life, for instance, in electrical power grids, oil and natural gas distribution, transportation systems, healthcare devices, household appliances, and many more.

Although there are many IoT application domains, we will address four of the main relevant ambient intelligence throughout this article, namely smart healthcare, smart transportation, smart cities, and smart manufacturing.

Smart healthcare and rescue systems have been improved by integrating information and communication systems into healthcare systems [6]. Smart healthcare enables Ambient Assisted Living (AAL) technology by adopting different sensors for health monitoring. The authors in [7] aim to advance the state of the art in evaluating and comparing AAL platforms and architectures by organizing an annual international competition. An important point about smart healthcare is that it needs a low-energy-consumption communication system and fast localization methods for the person(s) involved in the accident or emergency. IoT applications in smart transportation are mainly focused on parking systems, traffic, streetlights, congestion monitoring, and real-time public transportation [8]. Sensors in smart transportation (for example smart streetlights, magnetometers, and installed sensors in bikes and buses) provide the required information for effective system management. Smart transportation systems need long-range and low energy consumption communication systems and practical tracking methods. Smart cities benefit from provided information by massive densities sensors and actuators [9] to improve the quality of life, transport, traffic management, and interaction with the government. These sensors and actuators need communication technologies. The adopted communication technologies should provide communication with very low energy consumption over long distances. On the other hand, a smart city needs practical localization and tracking methods that lead to the effective use of sensor data. This can be useful to provide efficient services for IoT end nodes and efficient system management. Smart manufacturing is applied in industrial automation processes to improve monitoring and control, as well as, to increase the production efficiency in different industrial sectors (e.g., harbors, factories, etc.) [10]. As previously mentioned, long-range, low-power, and low-cost communication systems are essential in all categories of IoT-based systems.

Global Navigation Satellite Systems (GNSS) are the most commonly used methods for outdoor localization. These types of systems consist of constellations of geo-referenced high orbit artificial satellites that continuously transmit location and timing data. There are four main GNSS systems currently in operation: GPS (U.S.), GLONASS (Russia), Beidou (China), and Galileo (E.U.). These systems, typically, present accuracies in the magnitude order of 1 to 10 m, and have been used in IoT applications that demand higher accuracies and low device autonomy [11]. However, they present some important drawbacks that make them inappropriate for general IoT/IIoT outdoor localization: (1) they consume a lot of power (IoT systems have power battery limitation); (2) they are expensive (IoT localization methods should be low cost), and (3) they provide location information only for the IoT device itself, meaning that additional data must be forwarded to the gateways consuming additional power and bandwidth.

Based on range, network communication protocols can be divided into three groups: short-range (Bluetooth, RFID . . . ), medium-range (ZigBee, Wi-Fi, . . . ), and long-range (Low-Power Wide-Area Network (LPWAN)) [12]. LPWAN provides long-range communication among IoT devices with low-power consumption and low cost. Different technologies have been considered to be LPWAN, which include LTE-M, NB-IoT, Sigfox, and LoRaWAN [13]. LPWAN provides low-power, low-cost, and long-range communication and its signals can be used for communication and localization simultaneously. The LPWAN technologies

have star topology. Typically, IoT nodes send uplink transmissions to LPWAN gateways as end nodes. The LPWAN gateway sends collected data from end nodes to the LPWAN network server throw UDP/IP protocol. These data can be used for end-node localization by the LPWAN network server. LPWAN network server sends MAC control command and downlink packets to end nodes throw LPWAN gateways. LPWAN Application Servers can be owned by different organizations for distinct purposes.

There are different proposed localization methods that can be used to obtain the end nodes' relative location or for tracking purposes. Each category of these proposed methods has pros and cons that make them useful for specific IoT systems. Nevertheless, there are some limitations in proposed localization methods that need to be eliminated to meet the IoT ecosystem needs completely.

Obviously, there is an urgent need for the development of novel intelligent solutions to improve outdoor localization methods for IoT systems based on their limitations and requirements. This has motivated this paper to provide a survey paper to review, at first, different features and requirements of IoT systems on outdoor localization and, in the following, different proposed methods on each IoT system. Furthermore, this paper empowers researchers to create their own efficient and useful models for outdoor IoT localization by showcasing the state of the art employing LPWAN methods for location estimation. To the best of the author's knowledge, this is the first survey that reviews the latest research efforts focused on GNSS-free outdoor localization methods for the IoT ecosystem, having in mind its specific requirements and limitations. The main contributions of this work can be summarized as follows:

1. Definition of the main requirements and limitations of outdoor localization for the IoT ecosystem, having in mind the following application domains: smart healthcare; smart transportation; smart cities; smart manufacturing.
2. Review of the most relevant GNSS-free outdoor localization methods, with a focus on LPWAN technologies and their relationship with the newly 5G mobile network architectures;
3. Review of the most relevant methods for improving the localization accuracy in the IoT ecosystem.
4. Discussion on open challenges and future directions in the field.

This research has been conducted between May and August 2021. For LPWAN technology explanation, papers in the Google Scholar database from 2018 to August 2021 that contained the keyword "survey" and one of the keywords "LPWAN", "LoRaWAN", "Sigfox", or "NB-IoT" in the title were reviewed. For the location estimation method survey, there were around 2000 listed papers by Google Scholar database from 2018 to July 2021 that contained the keywords "LoRa", "Localization", and "outdoor". Due to many papers, we selected papers in three steps. At first, the most relevant studies were selected based on a manual title screening. Second, after the abstract screening, the papers with a proposed method or useful evaluation on outdoor localization are selected. Finally, the papers which provide a clear and smooth present of their subjects were reviewed.

The remainder of this document is organized as follows: Section 2 discusses the importance of localization in IoT ecosystem, their requirements, LPWAN technologies, and their architecture; Section 3 presents an overview of existent GNSS-free IoT Localization Methods and different proposed methods on localization accuracy improvement; Section 4 explained different IoT systems and relevant proposed localization methods; Lastly, in Section 5 the main conclusions, open issues, and future work guidelines are presented.

## 2. Outdoor Positioning in the IoT Ecosystem

### 2.1. Why IoT Location Matters?

With the entrance of IoT to different Industrial and non-industrial systems, it is predicted that the number of IoT end nodes will reach 50 billion by 2022. Therefore, the future of systems known as IoT systems is shaped by intelligent and moving IoT end nodes. Furthermore, IoT systems are moving toward autonomous operation with direct interaction

with the human, environment, and other systems [3]. For example, in an autonomous vehicle, the control system receives the achieved data of sensors as input. Based on receiving data, it sends appropriate vehicle manipulation commands (e.g., steering, accelerating, and braking) to actuators to achieve the goal travel path. To make this goal possible, it is important to achieve the location information of the vehicle. In addition, if the location estimation error becomes more than a specific threshold, it can make a wrong rotation decision. Wrong decisions in an autonomous vehicle can cause accidents and human harm. Accurate location estimation has become more critical in smart healthcare systems. In smart healthcare systems, the vital sign of use is determined using the adopted sensors. The measured vital signs are sent to the medical care center periodically. In an emergency, accurate location estimation of the user for faster and effective assistance is important. This issue becomes more complicated in Industry 4.0. The safe operation of Industry 4.0 is essential due to the widespread application of these systems in critical infrastructures. Autonomous systems in Industry 4.0 receive data from different mobile sensors which provide information about surroundings. Based on received data, control systems make a decision and sends control commands to actuators. In these systems, accurate location information of sensors and actuators is critical because decision errors in these systems can lead to catastrophic harm to humans, the environment, and other systems.

According to the mentioned points, to provide a safe IoT system, it is important to provide location information of IoT end nodes. Therefore, besides a robust and reliable communication network, it is important to provide an accurate location estimation method [4].

## 2.2. Application-Specific IoT Requirements

A localization system is composed of hardware and software components on both sides of the end node and network control. In the end-node side, LS can have size limitations. Therefore, it leads to choosing shorter antenna dimensions as radio frequency interfaces. By decreasing the antenna dimensions, path loss increases. As a result, the number of gateways on the receiver side in an invariant environment has increased to earn the same accuracy.

On the network control side, by increasing the number of end nodes, the volume of generated data increases. Therefore, a more powerful network control server is needed to handle data and earn required accuracy. According to the mentioned contents, different IoT applications based on their purpose have different localization system requirements [14]. In this section, important requirements for IoT localization system are identified and discussed.

- Security and Privacy: Long-range communication protocols provide connectivity for IoT systems and IoT systems provide information about the system, environment, and people. The information which is collected by IoT and IoT communication signals in LPWAN should be sent securely, otherwise, it can lead to security (disclose sensitive information about environment) and privacy problems (potentially disclose end-node location). Poor authentication and encryption methods can lead to end-node privacy violence [15].
  To provide a secure localization system, it is important to consider privacy and security requirements in each step of architecture. With the release of General Data Protection Regulation (GDPR), authors in [16] present an appropriate privacy-by-design to guarantee the GDPR compliance through the integration of a specialized GDPR-based access control system.
- Coverage Range: Different IoT systems work in different environment situations (for example, the smart industry has a harsh environment). LPWANs (LoRaWAN, Sigfox, and NB-IoT) has different coverage range and function in different situations. As a result, the efficiency and coverage range of the different LPWAN technologies can be varied in different systems. Therefore, a network technology should be compatible with the environmental situation and required range.

- Energy Consumption: Energy Consumption is one of the most important issues in IoT systems. IoT systems are embedded in different environments for an important purpose. Therefore, long battery life and the low energy consumption is a fundamental point for these systems. One of the important aspects to decrease energy consumption is multiple cheap network interfaces adoption such as Bluetooth, Wi-Fi, and LPWAN for short- and long-range positioning. In this case, end nodes look for nearby localization systems with the short-range network interfaces and, eventually, switch to a long-range search [17]. The authors in [18] try to standardize Application Programming Interfaces (APIs) designed to discover, access, and localize end nodes.
- Latency: low latency in some IoT categories is an essential priority because of the hard real-time limitations. In these cases, adopted LPWAN technology should provide services with specific maximum latency.
- Deployment: NB-IoT is a new emerging technology while LoRaWAN and Sigfox ecosystems are mature communication technology. Additionally, we need to choose the best LPWAN technology based on our system (local or public network).
- Accuracy: IoT systems can be divided into two categories based on required location accuracy: high accuracy and low accuracy. smart healthcare and smart factory need high accuracy based on their direct interaction with human safety. Smart cities and smart transportation need accuracies in the meter level. Therefore, the localization error rate of the adopted LPWAN method should be acceptable in the system.
- Scalability: LPWAN technologies are supposed to cover a wide-area environment. In different systems, a wide-area environment can contain many nodes that need to communicate throw a network and share the communication medium. For this reason, different technologies provide a range of communication options to handle network load to provide a scalable network. Therefore, the scalable of adopted technology should be acceptable for the system [19].

### 2.3. Location-Enabled IoT Technologies

This section gives an overview of different long-range communication, detailed covering the LoRaWAN layers and their attributes. In the following, the main features of 5G and the relation between LPWAN and 5G are explained.

#### 2.3.1. LoRaWAN

LoRaWAN's inventors aimed to develop a long-range, low-power technology to provide a scalable network for IoT systems. For this purpose, they used Chirp Spread Spectrum (CSS) modulation technology. CSS was widely used for sonars in the maritime industry and radar in aviation [20]. In CSS modulation, symbols are transmitted throw constantly increasing or decreasing frequency. CSS is known to provide resilience against interference, multipath, and Doppler effects. In February 2015, the LoRaWAN Alliance was founded, and the networking protocol was renamed "LoRaWAN". LoRaWAN is a Long-Range Wide-Area Network technology developed by Semtech. The LoRaWAN medium access control (MAC) protocol called LoRaMAC is an open-source protocol [21]. LoRaMAC defines different classes for end-users and security layers to enable secure bidirectional communication. Therefore, LoRaWAN provides the possibility for private network deployments by working in a license-free spectrum and its open access specifications. This subsection discusses different layers and features of LoRaWAN.

There are four key parameters to determine the CSS of the LoRaWAN physical layer. By incorporating these four parameters, LoRaWAN can work in different situations. each one of these four parameters can have unique variations. Four key parameters are channel, BandWidth (BW), spreading factor (SF), and transmission power (TX-PoW) [22].

- Channel: LoRaWAN works in license-free channel of 433 and 868 MHZ (in Europe), 915 MHZ (in US), and 430 MHZ (in AS) [23]. Lower Carrier Frequency leads to lower path loss. Therefore, Lower Carrier Frequency can be useful in long-distance. On the other hand, antenna dimensions at Lower Carrier Frequency should be bigger for a

stated radiation efficiency. It leads to inefficiency for IoT systems with size limitations. In addition, a narrower Carrier Frequency leads to a lower number of communication channels. As a result, it is not suitable for systems with many users.

- Spreading Factor (SF): Spreading technique leads to extend a symbol to a longer sequence of bits. This technique reduced the signal-to-noise and interference ratio. In LoRaWAN, symbols are encoded using several chirps. The number of chirps is determined by the spreading factor. SF in LoRaWAN can range from 7 to 12. Therefore, SF7 means that each chirp represents seven bits. According to the explanations, SF12 has the lowest data rate and highest energy consumption but it has the highest resilience against noise, interference, and other unpredictable time-varying impairments because of user mobility. Therefore, SF12 is useful for IoT systems that work in dynamic and harsh environments and SF7 is useful for IoT systems with power limitations.

- Bandwidth (BW): Bandwidth measures the amount of data that can be transmitted per unit of time. The duration of each chip is equal to $1/BW$. Therefore, the symbol duration ($T_s$) is equal to $2^{SF}/BW$. As a result, by increasing the BW, symbol duration and chirp duration would change accordingly which leads to a better Signal-to-Noise Ratio (SNR). Bandwidth in LoRaWAN can range from 7.8 kHz to 500 kHz.

- Transmission Power (TX-PoW): TX-Pow determines the direct amount of power used to transmit a chirp. By increasing TX-PoW, the probability of signal resistance versus channel fading increases too and there will be more chances for correct transmission of data. The LoRaMAC layer provides the medium access control mechanism. LoRaWAN has a star topology. Therefore, end nodes in LoRaWAN can only communicate with LoRaWAN gateways. For communication and data transmission, end nodes must join to network and their connection is allowed by the LoRaWAN network server. By allowing a LoRaWAN network server, end nodes will earn a set of parameters that are necessary to operate in a LoRaWAN. End nodes can join the network in two ways: Over-The-Air Activation and Activation by personalization [24]. In Over-The-Air Activation, the end nodes are identified with an end-device identifier, application identifier, and AppKey. In Activation by personalization, end nodes have two keys determined by the LoRaWAN network server as session keys. Therefore, in this way, the end nodes directly join the network without going through the Join procedure steps. After Joining the LoRaWAN network, end nodes can exchange data with the LoRaWAN network server throw LoRaWAN gateways. LoRaWAN defines three classes of end devices: Class A, Class B, and Class C [25].

  - *Class A*: when an end node needs to send data (Uplink), it selects a channel randomly. Then, end nodes send packets through the ALOHA technique. To enable bidirectional communication, Uplink transmission is followed by to short packet from the LoRaWAN network server (Downlink). Class A is useful for IoT devices with power limitations and low downlink data because the end node will be in sleep mode most of the time.
  - *Class B*: this class is useful for IoT devices with more downlink data and power limitations. The uplink transmission is the same as class A, but end nodes will open additional receive windows at scheduled times to receive downlink transmission. End node and LoRaWAN gateway establish a parameter. This parameter is used to calculate the ping slot. Throw ping slot determination, end device knows times that must wake up and wait for a downlink. Additionally, the server knows when end nodes are listening to the medium.
  - *Class C*: this class is not useful for devices with power limitation. In this class, IoT devices are continuously listening to the channel and are open for downlink transmission.

2.3.2. Sigfox

Sigfox is a wireless technology proposed for long-range and low-power communication for IoT systems. Sigfox works in a license-free spectrum, and it provides network

coverage in different countries by collaboration with different companies. Therefore, Sigfox does not allow private networks. This subsection discusses different layers and features of Sigfox. Sigfox provides unidirectional and bidirectional communication. It works in the license-free channel of 868 MHz (in Europe), 902 MHz (in the US), and 433 MHz (in Asia). Sigfox adopts Ultra Narrow Band (UNB) radio transmission for both uplink and downlink communication [26]. UNB provides long-range communication by passing through solid objects. The uplink bandwidth of Sigfox is 100 Hz (in Europe) and 600 Hz (in the US). The downlink bandwidth of Sigfox is 1.5 kHz. Sigfox uses Differential Binary Phase-Shift Keying (DBPSK) modulation for uplink transmission and Gaussian Frequency-Shift Keying (GFSK) modulation for downlink transmission. DBPSK has more efficiency in the spectrum medium access than GFSK that leads to uplink range increment. By doing this, the lower transmit power in the uplink band is compensated [27].

Similar to LoRaWAN, Sigfox has star topology and asynchronous communication [28]. Asynchronous communication allows the end nodes to be in sleep time most of the time. Sigfox Architecture is similar to LoRaWAN. End nodes are connected to base stations, and they can communicate with the Sigfox network server throw base stations. Connection to several base stations leads to handover protocol needless. Sigfox network server connect to base stations throw IP/UDP network. Sigfox network server collects data from base stations, and it is responsible for network control and end nods authority.

Sigfox MAC employs slotted ALOHA protocol and provides two types of communication between an end node and base station: unidirectional and bidirectional transaction. In the unidirectional transaction, the end-node transmits several uplink frames in a randomly selected frequency channel and at different time intervals. This leads to resilience in the presence of issues such as interference and multipath fading. In a unidirectional transaction, the Sigfox network server cannot send downlink data to the end nodes. In a bidirectional transaction, after the end node sends uplink data same as a unidirectional transaction, the Sigfox network server sends downlink data to the end node.

### 2.3.3. NB-IoT

Narrow Band IoT (NB-IoT) has introduced by the 3rd Generation Partnership Project (3GPP) in 2016. On the contrary to Sigfox and LoRaWAN, NB-IoT operates in licensed spectrum and synchronous communication. Therefore, it provides higher traffic reliability and is preferred for IoT systems that need guaranteed quality of service (QoS). The NB-IoT communication protocol is based on LTE and it reduced its power consumption by reducing LTE protocol functionalities. NB-IoT is compatible with traditional cellular networks. Therefore, it can work in LTE or GSM under licensed frequency bands. NB-IoT has a frequency bandwidth of 200 kHz, and it uses Orthogonal Frequency Division Multiple Access (OFDMA) for downlink and Single Carrier Frequency Division Multiple Access (SC-FDMA) implemented for uplink communication. It leads to a 250-kbps data rate for downlink and a 20 kbps data rate for uplink. NB-IoT uses LTE protocol to let end nodes join the network and connect to one station. Therefore, NB-IoT needs to hand over protocols. As NB-IoT is synchronous communication, end nodes consume more energy than LoRaWAN and Sigfox.

### 2.3.4. LTE-M

To provide cellular connectivity for machine type communication, Long-Term Evolution for Machines (LTE-M) was introduced by 3GPP [29]. Like NB-IoT, LTE-M operates in licensed LTE spectrum and uses synchronous communication for QoS guarantees. LTE-M can work with a bandwidth that is equal to a single GSM channel (200 kHz). The most relevant characteristic of LTE-M is low cost, specifically on the IoT device side, due to the possibility of using less expensive radio modules [30]. Compared with NB-IoT, LTE-M supports up to 1 Mbps in uplink and up to 384 kbps in downlink. In addition, LTE-M has lower latency, but it typically consumes more power, leading to the decrease of the battery

lifetime [31]. LTE-M is compatible with LTE or GSM, and it has a similar medium access control mechanism.

### 2.3.5. 5G

The fifth-generation cellular network (5G) is a response to the IoT requirements caused by continuous expansions of these systems including massive connectivity, security, trustworthiness, coverage of wireless communication, ultra-low latency, throughput, and ultra-reliable. Several key techniques enable 5G to meet the IoT requirements. These key enabling techniques are Network Function Virtualisation (NFV), Heterogeneous Networks (HetNet), Software Defined Networking (SDN), Massive MIMO, and Beamforming techniques. To handle IoT systems that need long-range communication with low power consumption and low bandwidth, 5G will adopt LPWAN technologies. In [32], Li et al. introduce three main 5G application categories include Ultra-Reliable and Low Latency Communication (URLLC), Enhanced Mobile BroadBand (eMBB), and massive Machine-Type Communication (mMTC). They mentioned that LPWAN can provide an important application direction for 5G, especially mMTC, applications. On the other hand, they stated that 5G can help the development of LPWAN by building the fundamental infrastructure for communication services. In [33], the authors investigate the possibility to seamlessly integrate LPWANs with the upcoming 5G technology. They identified and discussed the four possibilities of integrating the LPWAN with a cellular network. The first two options are based on the use of either a 3GPP access network or an untrusted non-3GPP access network for the LPWAN gateway. The latter two options imply merging the LPWAN gateway with either the eNodeB or integrating it with the core network. The pros and cons of these solutions have been highlighted and discussed in the paper. Therefore, it is necessary to implement a transparent integration between LPWAN technologies with the upcoming 5G. Table 1 presents a summary of the most relevant LPWAN technologies. In addition, a graphical comparison of the main properties of LPWAN technologies is depicted in Figure 1. In this figure, the scalability stands for the number of supported devices (NB-IoT offers a very high scalability reaching 100 k end nodes per cell, while Sigfox and LoRaWAN can support up to 50 k end nodes per cell). Range determines the maximum distance that can be covered by an antenna cell (Sigfox covers a larger environment than LoRaWAN and NB-IoT). A mature technology will become established and commercially viable when adopted on a global scale, increasing the technology's trustability. The deployment factor identifies the time taken by a specific technology to become mature, meaning that the recent worldwide popularity of the LoRaWAN protocol results from its trustable and mature communication technology. Battery life specifies the average autonomy of the system (LoRaWAN is more energy efficient). Cost efficiency of a technology is determined based on various aspects including spectrum cost (license), network/deployment cost, and device cost (Sigfox and LoRaWAN are more cost-efficient when compared to NB-IoT). As NB-IoT works in a licensed spectrum, it can guarantee the QoS. Finally, NB-IoT and LoRaWAN (by providing class C) offer lower latency communication than Sigfox.

As introduced earlier, long-range, low-power, and low-cost communication systems are crucial for the success of the IoT deployments. As depicted in Figure 1, LoRaWAN and Sigfox are more appropriate for IoT deployment due to their higher cost efficiency. Additionally, LoRaWAN technology has important strengths that may include: supporting private network deployment, no restriction on the number of messages per day, supporting delay-sensitive systems using different classes, providing more information about the channel because of the larger message payload size, and more efficient power consumption. As a result of these characteristics, LoRaWAN technology has become a widely used communication technology in several IoT application domains, and this article focuses the exploration of low-cost GNSS-free outdoor localization techniques with a focus on the LoRaWAN technology.

**Table 1.** Comparison of main technical features of LPWAN technologies.

| Technology | LTE-M | NB-IoT | Sigfox | LoRaWAN |
|---|---|---|---|---|
| Frequency Channel | Licensed LTE frequency spectrum | Licensed LTE frequency spectrum | License-free channel of 868 MHz (in Europe), 902 MHz (in US), and 433 MHz (in Asia) | License-free channel of 868 MHz (in Europe), 902 MHz (in US), and 433 MHz (in Asia) |
| Bandwidth | 200 kHz | 200 kHz | 100 Hz (in Europe) and 600 Hz (in US) for uplink and 1.5 kHz for downlink | Range from 7.8 kHz to 500 kHz. |
| Modulation | QAM | QPSK | DBPSK modulation for uplink and GFSK modulation for downlink | CSS (Fsk is defined too) |
| Maximum Throughput | 1 Mbps | 200 kbps | 100 bps | 37.5 kbps |
| Maximum Range | 5 km | 1 km (in urban) and 10 km (in rural) | 10 km (in urban) and 40 km (in rural) | 5 km (in urban) and 20 km (in rural) |
| Packet Payload size | 100 bytes | 1600 bytes | 12 (uplink packet) and 8 (downlink packet) bytes | 243 bytes |
| Battery life time (1000 mAH) | 9 month | 3.5 years | 4.5 years | 5 years |
| Maximum message per day | Unlimited | Unlimited | 140 message (uplink) and 4 message (downlink) | Unlimited |
| Topology | Cellular Network | Cellular Network | Star | Star |
| Authentication | LTE encryption | LTE encryption | Not supported | Over-The-Air Activation and Activation by personalization |

**Table 1.** *Cont.*

| Technology | LTE-M | NB-IoT | Sigfox | LoRaWAN |
|---|---|---|---|---|
| Advantage | Higher traffic reliability with low delay | Higher traffic reliability | Wide communication range | Allow private network construction, Provide energy consumption Management by different class type definition, Adaptive data rate |
| Disadvantage | Higher energy consumption (more than NB-IOT) | Higher energy consumption | Do not allow private network, Has a maximum message limitation per day | signal attenuation and interference |

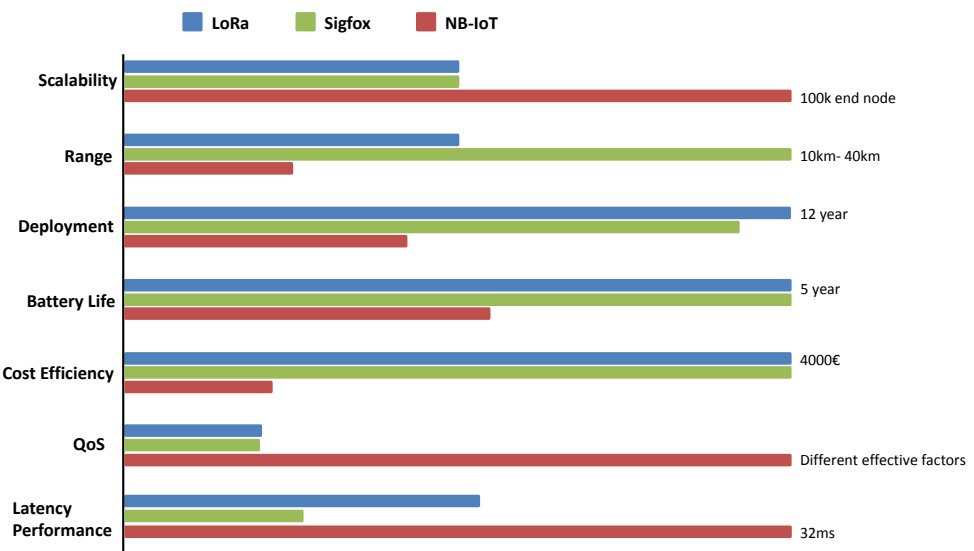

**Figure 1.** Performance comparison of LPWAN technologies. Adapted from [34].

*2.4. Common LoRaWAN-Based Localization Architectures*

A LoRaWAN-based localization system is composed of hardware and software in both the user and network control sides. User side components include, one/several sensor elements, a processing unit, a radio frequency module, an antenna etc., and the network control side typically includes a processing unit, networking unit, power source system, etc. The user agent on behalf of the end node starts sending data to the network control sides. The data transmission frequency of the end node varies in depending on the IoT application domain. Network control system determines the end-node location based on received data. On the other hand, in a large area, several service providers can deploy different methods for end-node localization and different hardware and processing techniques may be considered. Network control system must enable the cooperation between different methods and their functional modules [17,35].

In a large area, by increasing the number of end nodes, the variety of localization systems and the number of transmitted packets to the network control system may increase, leading to a latency increment and, as a result, a performance decrease on the localization accuracy.

Currently, two distinct localization architectures have been proposed for the IoT ecosystem: Cloud-based and Edge-AI-based. In this subsection, these architectures are explained.

2.4.1. Central Architecture (Cloud-Based)

Cloud computing emerges as the key platform for localization system data storage, processing, and analytics due to its simplicity, availability, and scalability. In cloud-based localization systems, when gateways receive data from end nodes, they send data to a network server. Network servers aggregate and forward data to the cloud layer through the Internet. Cloud layer, at first, discovers the adopted method by system and user agent for end-node localization. Then, the end-node localization and user interfaces are provided. Cloud-based method decreases the localization latency by providing storage, computational, and processing power. Therefore, received data will be accessible for real-time positioning and monitoring. Cloud-based architecture is shown in Figure 2.

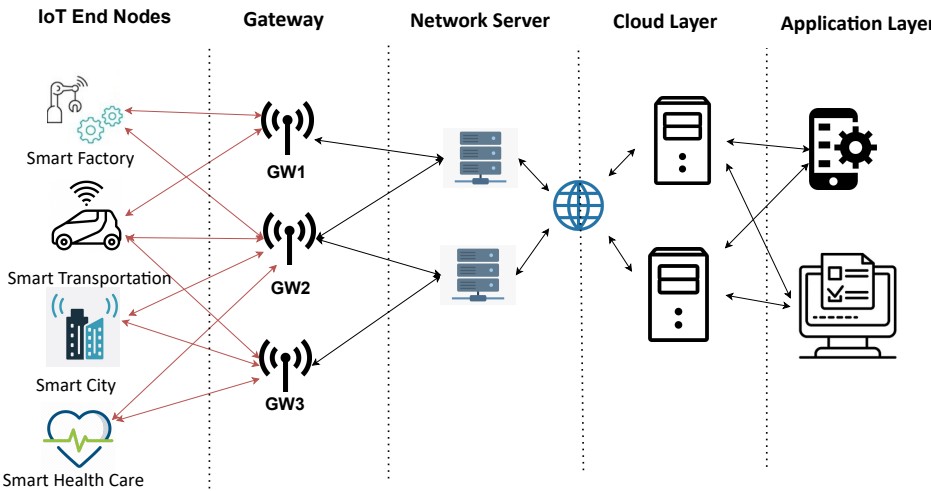

**Figure 2.** Cloud-based architecture.

### 2.4.2. Distributed Architecture (Fog and Edge-AI Based)

With increasing the need for real-time processing and computing, fog computing appeared as a suitable solution. In Fog and Edge-AI-based architecture, the gateway layer act as a smart edge layer [36,37]. At the smart edge layer, artificial intelligence algorithms are applied to enhance the quality of service (QoS). For example, different gateways can work in distinct environmental situations. Based on each particular environment, each edge gateway can determine environment error sources and reduce their effects on received signals. On the other side, some prepossessing (e.g., the adopted method by system and user agent for end-node localization discovering) and important evaluation (distance estimation) on receive data can be done on the edge layer. This mechanism helps to reduce latency (especially in emergency systems such as smart healthcare systems) and improve the overall system performance. As a result, location estimation error decreased. Then, the results are sent to the network server as a fog layer for aggregation and more advanced processing and forwarding to the cloud layer through the Internet. This method leads to efficient use of network bandwidths. Additionally, by increasing the number of functions and the number of connected devices to the fog and edge Layers, an additional orchestration layer is required for management purposes. The authors in [38] provide thorough functional and performance comparison with container orchestrators (in particular Docker Swarm and Kubernetes) under different real topologies in the fog-based cluster, using wired and wireless infrastructures. An example of an Edge-AI-based architecture is depicted in Figure 3.

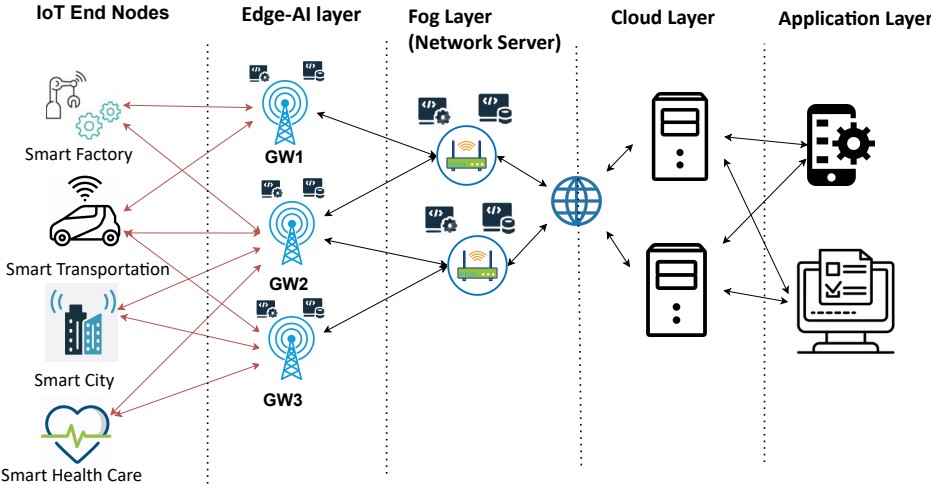

**Figure 3.** Fog and Edge-AI-based Architecture.

### 3. GNSS-Free IoT Localization Methods

End-node devices are the core of IoT systems. End nodes provide the required information for the proper operation of the IoT system. For effective use of provided data, end-node location determination is essential. On the other hand, to provide appropriate location-based service for end nodes, the LPWAN network server should be aware of end nodes' location. According to the explanation, IoT localization is a key requirement for system performance. In this section, localization methods are expressed in two general categories: (1) Signal-Based localization methods and (2) Learning-Based localization methods.

In Signal-Based localization methods, Time-Based localization methods provide more accuracy, but they lead to extra cost. Learning-Based localization methods provide more reliable estimation in the presence of environment error sources. The best localization method should be adopted to make a good trade-off between achieved performance, time, and cost [39]. A large category of studies on outdoor location estimation works on accuracy improvement regardless of system category. In the following section, we describe some of these proposed methods.

#### 3.1. Signal-Based Localization Methods

In this category, LPWAN signal features of IoT are being used for end-node localization. Power-based (RSSI), time-based (ToA, TDoA), and angle-based (AoA) methods are the most important signal characteristics [40].

- Received Signal Strength Indicator (RSSI): RSSI is the simplest and widely used method that uses received signal power for end-node localization. For doing this, a path-loss model should be developed for the system. The path-loss model specifies the signal propagation model of the system. The signal propagation model determines the signal strength at increasing distances [41].

  There are different kinds of propagation models including Line Of Sight (LOS), Two-Ray (2R), Nakagami-m, gamma distribution, and log-normal, etc. In the simplest model, a free space between transmitting and receiving antennas is considered. Therefore, only a LOS propagation path is present between transmitter and receiver antennas. In this system, the only disturbance is atmospheric effects and path loss is a function of distance between them. In more usual systems, communication between transmitter and receiver takes place closer to the earth surface. Therefore, the LOS model is extended to incorporate the effect of the ground-reflected signal. That means, in the 2R model, the path-loss characteristics in LOS environments are dominated by interference between the direct path and the ground-reflected path [18].

  In a more realistic system, the transmitted signal is obstructed by buildings, mountains, and foliage several times and a combination of these signals is received. Received signal strength in multipath propagation at the receiver can be characterized based on two different scales (large scale and small scale). Large-scale fading is due to motion in a large area, and small-scale fading is due to small changes in position or to changes in the environment [42]. To model these two different scales of multipath propagation, different probability distributions are presented (e.g., Nakagami-m, gamma distribution, log-normal). As most of the IoT systems have small changes in position or in the environment (smart healthcare, smart manufacturing, and most applications in smart city), in this paper we focused on small-scale fading to model the signal propagation.

  The log-normal model or log-distance path-loss model is the most commonly used method to model small-scale fading. In this model, when the LoRaWAN network server receives a signal from the end node, it can measure transmitter distance by Equation (1).

$$RSSI = -10n log_{10}(d) + A \qquad (1)$$

  where $n$ is the path-loss exponent and $A$ is the RSSI value at a reference distance (1 m) from the receiver. The ground reflection model is another commonly used propagation

model. The ground reflection model considers the ground reflection propagation path between transmitter and receiver in addition to the direct path [43]. RSSI is easily affected by the propagation environment, absorption and scattering, and antenna effects including impedance mismatch and polarization mismatch [44]. In addition, RSSI is highly dependent on changes in environmental and weather conditions, which may lead to location accuracy reduction. On the other hand, RSSI is cost-effective and it does not need extra equipment.

- Time of Arrival or Time on Air (ToA): ToA is the time that transmitted signal takes to reach the receiver and the ToA of a packet is dependent on CR, SF, and BW. ToA can be used for distance estimation by multiplying the speed of light. There are two categories of ToA: one-way and two-way ToA [45]. In one-way ToA, the distance is computed based on one-way signal transmission between transmitter and receiver. In this case, ToA requires strict synchronization or timestamps to be transmitted with the signal. In two-way ToA, ToA is computed from roundtrip communications between transmitter and receiver. Two-way ToA is an asynchronous procedure, thus involved clocks do not need to be precisely synchronized.

- Time Difference of Arrival (TDoA): unlike ToA, TDoA needs three or more GWs with only precise time synchronization between GWs for end-node localization. When an end node in location $(x,y)$ transmits a message, the message is received by gateways with different timestamps $(t_1, t_2, \ldots, t_n)$. Then, differences in signals propagation times $(t_{i,j} = t_i - t_j)$ is measured at the receiver for physical distance computation by Equation (2).

- Angle of Arrival (AoA): In the AoA method, an array of antennas is implemented at the receiver side to estimate the angle between transceiver and receiver. For doing this, the time difference of transmitted signal at individual elements of the antennas array is computed. The AoA method is not dependent on signal strength for location estimation. Therefore, channel fading and environment features have minimal effect on location estimation accuracy. In addition, AoA can decompose the received signals into direct and reflected paths, thus, the multipath effect can be minimized [46]. AoA has a high deployment cost, and it needs careful calibration. A low error in angle computation can lead to huge error in location estimation.

$$t_{i,j}.c = \sqrt{(x_i - x)^2 + (y_i - y)^2 + (z_i - z)^2} - \sqrt{(x_j - x)^2 + (y_j - y)^2 + (z_j - z)^2} \quad (2)$$

As TDoA work with propagation time, it is dependent on CR, SF, and BW of the system same as ToA.

### 3.2. Learning-Based Localization Methods

As mentioned, each category of Signal-Based localization methods has pros and cons that make them suitable for a special system. However, all these categories are implemented for a definite environment. Therefore, they become unreliable in the presence of environment error sources that are not included in the underlying models. To overcome this issue, Machine Learning (ML)-based localization methods are becoming increasingly popular [47]. ML methods are used in combination with Signal-Based localization methods. In this paper, Fingerprinting and ML methods for Path-Loss modeling have been investigated.

- Path-Loss Modeling: When a signal travels from the transmitter to receiver over a distance $d$, path loss or propagation loss leads to received signal power decrements. By correct determination of system path-loss model, location estimation accuracy improved. The path-loss model of the signal can change based on the environment. For path-loss model determination, the ML method needs collected data of experiment environment under different situations by Equation (3).

$$PL = P_{T_x} + G_{T_x} + G_{R_x} + 10.log_{10}(1 + 1/SNR) - RSSI \quad (3)$$

where $P_{T_x}$ is the transmission power, $G_{T_x}$ and $G_{R_x}$ are the gains of the transmitting and receiving antenna, respectively. By large enough number of *SNR* and *RSSI* measurements as input of ML, correct determination of system path-loss model as ML output can be achieved. Free space, indoor and urban path-loss models are more usual defined path-loss models [48]. The authors in [49] studies the propagation of LoRaWAN signals in forest, urban, and suburban vehicular environments.

- Fingerprinting/Scene Analysis: The purpose of the fingerprinting method is to find the location of an end node by comparing its signal characterizes received from multiple GWs with a stored signal characterizes. For example, RSSI fingerprinting-based localization method has two phases: offline and online phase. In the offline phase, the characteristics of signals for specific locations (x,y) are collected and stored according to Equation (4).

$$F_t(x,y) = [RSSI_{1,t}, RSSI_{2,t}, \ldots, RSSI_{N,t}] \tag{4}$$

where $N$ is the number of gateways and $t$ is the time that signal is collected by gateways. In the online phase, the distance matching of received signal characterizes with stored data is used for location estimation. Distance matching indicates how close a fingerprint at one reference location to the received fingerprint at an unknown location [50]. The fingerprint process is shown in Figure 4.

- Proximity Analysis: In this category, service provider do not need precise location information of end user. Distance between two end user is the purpose of Proximity Analysis. This analysis is useful in smart factories [51].

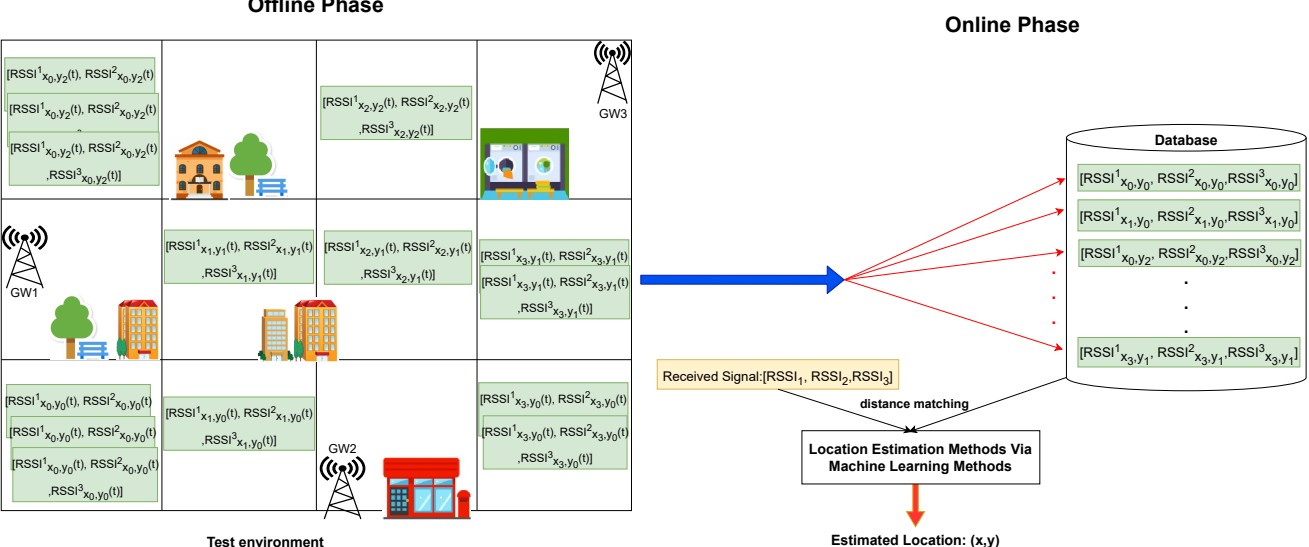

**Figure 4.** Fingerprint-based localization phases.

### 3.3. Localization Accuracy Improvement Methods

There is a large amount of literature that review different aspect of LPWAN technologies and Localization methods [41,52–54]. The authors in [52] review the performance of LPWAN from various aspects (signal propagation, coverage, and energy conservation). The authors in [41] characterized and implemented an outdoor LoRa-based localization system. The authors in [53] compares the location estimation accuracy of TDoA-based and RSSI-based methods in a public outdoor LoRa network. The authors in [54] investigate the efficiency of fingerprinting using RSSI for location estimation.

During the work with RSSI, data preprocessing and the hyperparameter tuning steps can optimize the performance of a localization system. The authors in [55] present a detailed study of four alternative methods of preprocessing (positive values, normalized

data, exponential, and powered) and discuss the process of selecting the most appropriate preprocessing methodologies and performing hyperparameter tuning.

RSSI-based methods provide acceptable location estimation for the indoor environment. However, they are negatively affected by unknown propagation in outdoor environments. Therefore, This leads to errors up to hundreds of meters. Different methods are proposed to improve the performance of RSSI-based localization by filtering out noisy measurements (filtering techniques e.g., Particle, Bayesian, and Kalman [43,56]) to improve accuracy.

The authors in [57] try to improve the distance estimation accuracy using a Wiener filter. The proposed Wiener algorithm combines the RSSI values received from different LoRaWAN modulation (10 different modes) configurations and formulate the distance logarithm as a linear combination of RSSI measurements. In addition, the authors mentioned that using only one mode, the proposed method has better estimation yet.

In addition, the authors in [58] proposed the Gaussian Signal Attenuation Model (GSAM) to decrease the interference effect in dynamic environments for vehicle localization. They combine the motion of vehicles to satisfy environmental constraints to improve the accuracy of RSSI-based localization.

Blocking or multipath fading in RSSI-based methods can lead to non-Gaussian noise. This non-Gaussian noise leads to an attenuation signal. The weak signal received by the gateway leads to location estimation error. Many different studies try to improve the location estimation by selecting the GWs that are not affected by non-Gaussian noise [59–62]. The authors in [59] proposed six new RSSI-based localization algorithms to reduce the effect of non-Gaussian noise in LoRaWAN networks. The first two algorithms eliminate GWs that are highly affected by non-Gaussian noise (RSSI-based LoRaWAN Localization with K-mean Clustering, RSSI-based LoRaWAN Localization with Iterative Elimination), the next two algorithms are proposed to select GWs that are not or are only slightly affected by non-Gaussian noise (RSSI-based LoRaWAN Localization with Minimum Mean Backward RSSI (MBRE) Error, RSSI-based LoRaWAN Localization with Density-based Clustering), and the last two algorithms make use of the previous algorithms to select a set of GWs that are not or are only slightly affected by non-Gaussian noise (RSSI-based LoRaWAN Localization with Minimum MBREs of Selected anchor nodes, RSSI-based LoRaWAN Localization with the Centroid of Selected estimated target locations).

For location estimation using RSSI, different ML methods are proposed. ML methods are divided into three categories: supervised (learning phase uses from labeled data), semi-supervised (learning phase uses from a small amount of labeled data and a large amount of unlabeled data), and unsupervised learning (learning phase uses from unlabeled data). In [63], the authors adopt Neural Network (NN) to estimate the location of an object (supervised learning). The authors use from provided data by [23] for simulation. They provide their simulation in three models: unprocessed data, processed data, and processed data using mean of measurements. During the simulation, authors change different factors of NN (number of hidden layers, number of neurons in each layer, optimizer, earning rate, the batch size, and the number of neurons) to achieve the best results. The authors mentioned that NN can gain equally or even better estimation compared to kNN fingerprinting.

Deep learning or deep neural network (DNN) is the extended model of artificial neural networks to learn tasks. It contains more hidden layers. There are large category of literature that adopt DNN for location estimation [64–66]. The authors in [67,68] mentioned that it is not feasible to work only with labeled samples in an outdoor environment. In addition, the location estimation accuracy of the semi-supervised learning methods is usually better than that of unsupervised learning. Therefore, they proposed a semi-supervised DNN for location estimation in LoRaWAN. By constructing the relationship between labeled and unlabeled data, authors generate several virtual labeled data through learning. RSSI, SNR, and timestamps are used for training and learn grid segmentation knowledge from the source domain and transfer the knowledge to the target domain.

RSSI-based outdoor location estimation methods have acceptable performance in certain environments and their error increased in the general environment. The authors in [69] proposed to exploit from UAVs. The proposed method used from a UAV for localization estimation improvement was initially provided by the network. The authors characterize the relevant parameters of the communication system and used it from a quadrotor capable of carrying the required hardware as a mobile gateway. Their result showed that the mobile gateway (between 1 and 3) can lead to localization precision increment.

RSS-based fingerprinting approaches can usually achieve better accuracy than RSSI-based methods. This makes fingerprinting methods an interesting option for location estimation. However, Fingerprint-based methods require a wide survey of the environment to build a database and update the database regularly to reflect changes in dynamic environments and complex signal fading of different environment situations (e.g., weather changes).

The authors in [54] analyze RSSI fingerprinting in real LOS and non-LOS (NLOS) environments. The authors try to determine the appropriate function for accurate LoRaWAN packet specifications and RSSI mapping. They examine the performance of LoRaWAN at different spreading factors and showed that ML algorithms could use RSSI to predict the number of obstacles.

In [70], authors mentioned that location estimation using the traditional path-loss models cannot be determined because of highly variable factors. Therefore, they studied various regression (linear, polynomial, exponential, and Gaussian regression) and machine learning models (support vector machines, spline models, decision trees, and ensemble learning) on received RSSI to accommodate the variability of the ranging function. The authors point out that the dependence of gateways on the environment (e.g., weather conditions and obstacles) greatly influence signal quality and can be accommodated by training different models for each gateway.

In addition to the different environment influence on gateways, localization errors can vary in different environmental and different gateways. The authors in [71] mentioned that localization errors are considerably larger at the edges of an environment, in comparison to its center. As a result, the localization error of a new location estimate is associated with the localization error of its nearest ground truth location(s). Therefore, the aggregate statistical metrics are intrinsically not very useful in characterizing the localization error of a single location estimate. In summary, they proposed to use a trained model to estimate the localization errors at new locations based solely on the observed RSS values at these locations and they used estimated location as an optional additional input feature for different regression methods (ordinary least squares, ridge, lasso, elasticNet, polynomial, k-nearest neighbors, support vector, and random forest).

Path-loss parameters are different in various environments. These different parameters lead to distinct RSSI for the same distance. The authors in [72] proposed to use high-resolution satellite images to generate virtual fingerprints for land-cover types and path-loss parameters identification for each gateway. In the training phase, a random forest model for automatically generating a land-cover map according to the satellite image of the area of interest is used. Next, the proposed method produces a virtual fingerprinting map. Then in the localization phase, they combine the fingerprinting maps of multiple gateways for node localization.

In [73], authors mentioned that in RSSI signals cannot be measured for every location in the service area. Therefore, they proposed to estimate the locations using probabilistic means based on three different algorithms that use interpolated fingerprint RSSI maps.

Time-based approaches in LoRaWAN lead to more accurate location estimation. The authors in [53] assess the RSS positioning performance in a realistic LoRaWAN network, for different spreading factors (SFs), and make the comparison with LoRaWAN TDoA performance. In their experiments, they pointed out that raw location estimates of the TDoA approach result in better accuracy than all investigated RSSI approaches with different SF. In

the following, the authors improve the performance of RSSI-based methods by taking into account road-mapping filter (the road infrastructure layout and the allowed speed limits).

In [74], the authors proposed a TDoA-based method for localization that considers road map and movement speed. They determine the positioning accuracy for three different mobility profiles (walking, cycling, and driving).

The authors in [75] divide TDoA localization algorithms into two categories (Non-iterative and iterative algorithms). They explain 5 classic TDoA algorithms and evaluate their performance using simulation. Based on the results, they select the two providing the best accuracy (i.e., Chan's and Foy's) and combine them to improve localization accuracy.

One of the most important drawbacks of TDoA-based methods is their need for time synchronization. The authors in [76] proposed an indoor localization system based on TDoA for UWB. The proposed method relies on the capability of the IEEE 802.15.4a UWB receiver for accurate time distinctions of the transmitter. In [77], Lopes et al. propose a TDoA-based method for centimeter-level indoor localization of smartphones using a WSN infrastructure and non-invasive audio for TDoA estimation.

The authors in [78] mentioned that in some harsh environments LoRaWAN cannot work acceptably so it needs an extra assistant. They adopt ultrasonic signals as an aiding approach for TDOA localization-based method for outdoor environments.

For localization accuracy improvement, The authors in [79] proposed a fingerprinting method applied to a reference map to perform outdoor geolocation based on machine learning (Random Forest and Neural Networks). The map combines TDOA measurements.

There is a category of literates that uses a combination of two localization methods to improve estimation accuracy [80–83]. The authors in [80] present a message-passing algorithm for target tracking which exploits AoA and ranges information from RSSI measurements. In the proposed method, the path-loss exponent to each anchor is assumed to be unknown and time-varying to be adaptive to dynamic propagation conditions.

In [81], RSSI and ToA are integrated for the regression analysis of the distance between the anchor node and the IoT node. The proposed method is restricted to localizing a single target. To solve this restriction, the authors in [82] proposed a new RSSI and ToA-based localization method. Table 2 provides a summary of different localization methods.

**Table 2.** Summary of proposed accuracy improvement methods for localization regardless of system category.

| Reference | Localization Technique | Method |
| --- | --- | --- |
| Savazzi et al. (2019) [57] | RSSI | Wiener Filter for noise decrement |
| Biswas et al. (2021) [58] | RSSI | GSAM to decrease interference effect |
| Lam et al. (2019) [59] | RSSI | Six algorithm for non-Gaussian noise decrement |
| Daramouskas et al. (2019) [63] | RSSI | NN for Location estimation |
| Chen et al. (2019) [67] | RSSI | DNN for location estimation |
| Chen et al. (2021) [68] | RSSI | DNN for location estimation |
| Delafontaine et al. (2020) [69] | RSSI | Exploit from UAV for accuracy improvement |
| Anjum et al. (2019) [54] | RSSI-based Fingerprint | Analysis of regression and ML models |
| Lin et al. (2020) [72] | RSSI-based Fingerprint | High-resolution satellite images adoption |
| Choi et al. (2018) [73] | RSSI-based Fingerprint | Use interpolated fingerprint map |
| Podevijn et al. (2018) [74] | TDoA | Road map and movement speed consideration |
| Pospisil et al. (2020) [75] | TDoA | Combine Chan's and Foy's TDoA algorithms |
| Elsabaa et al. (2019) [78] | TDoA | adopts ultrasonic signals |
| Carrino et al. (2019) [79] | TDoA-based fingerprint | Random Forest and NN |
| Lin et al. (2020) [44] | AoA and RSSI | Dynamic path-loss exponent consideration |
| Hu et al. (2019) [81] | RSSI and ToA | Regression analysis |
| Chen et al. (2020) [82] | RSSI and ToA | Regression analysis |

## 4. LoRaWAN-Based Outdoor Positioning in Application-Specific IoT Domains

In this paper, four IoT application domains are considered and discussed. There are several proposed methods based on LoRaWAN location estimation for IoT systems. The proposed methods intend to meet the limitation of IoT systems. In this section, the requirement and limitations of these basic IoT systems are explained. In the following, we overview some of the proposed localization methods for each application domain.

### 4.1. Smart Manufacture (Industry 4.0)

The emergence and integration of new technologies (wireless sensors networks, IoT, Cloud computing, and big data) with industries lead to Industry 4.0 introduction by German during the Hannover Fair event in 2011 [84–86]. The purpose of Industry 4.0 is efficiency and safety increment. For this purpose, four principles are mentioned: (1) Interoperability (all devices in the industry become connected and communicate with each other), (2) Information transparency (empowering industries with different sensors to create a virtual copy of system conditions), (3) Technical assistance (providing operations to support human decisions), and (4) Decentralized decisions (enabling autonomous machines for making efficient decisions). According to the explanations, Industry 4.0 needs a long-range communication technology to provide connectivity for sensors (principle 1). For correct condition monitoring, Industry 4.0 needs high-accuracy localization methods. High accuracy localization leads to have a better awareness of environmental conditions and make the best decision (principles 2 and 3). Based on this, LoRaWAN is a good option to provide communication and localization technology. The authors in [87] mentioned the main problems of usual wireless communication systems (small distance limitations of wireless communication, weak security or complete lack of security in data transmission, and high energy consumption). To address these limitations, they propose, implement, trial-run, and benchmark a LoRa-based flexible hardware architecture for use in industrial remote monitoring and control. The authors in [88] designed, developed, and tested a smart automation system based on LoRaWAN for easy monitoring and controlling of the appliances. A new LoRaMAC layer is proposed in [89] to provide the scheduling of real-time data flow.

Localization Methods in Industry 4.0

The smart industry is composed of a wide variety of sensors and actuators operating in a widespread area. LoRaWAN provides an effective communication network to locate, monitor, and control devices in a long-range area. Location estimation based on LoRaWAN is used in different sectors of industry (factory, agriculture, harbor, etc.). In this subsection, proposed methods for different industry sectors are discussed.

The authors in [90] made a comparison about the performance of LoRaWAN and technologies used for industrial asset identification, assessment, localization, and tracking (Radio Frequency Identification (RFID) and Near Field Communication(NFC)) and their strengths and weaknesses. They mentioned that RFID and NFC fall short when compared to LoRaWAN technologies. The authors in [88] demonstrate that a LoRaWAN base system for temperature and humidity monitoring of an industrial environment can detect fire with an accuracy of 90% and control the switching functionality with 92.33% accuracy. The authors in [91] discussed the characteristics and behavior of LoRaWAN in monitoring the industrial environment.

It is predicted that almost 75% of the world's population will live in urban areas by 2050. By increasing the people's residence in urban areas, it is expected that the food production amount increased to double. As a result, national economies try to develop agriculture by emerging technology employment. The authors in [92] discussed how the combination of IoT and data analysis is enabling smart agriculture and review several benefits and challenges of them. Sensors and robots play an important role in smart agriculture. The sensor is used to gather weather and situation information and robots employs to solve a problem such as field activities, crop harvesting, livestock breeding,

monitoring of agricultural areas, etc. LoRaWAN is an effective and useful wireless network to provide communication between sensors and agriculture servers. The authors in [93] mentioned that localization of the sensor nodes in agriculture is also an important factor for identifying a node located within the large area of an agricultural field and discusses the role of localization for deploying the sensor nodes in agricultural fields. They proposed an IoT-based WSN architecture for real-time monitoring of agricultural fields and they implemented an energy harvesting mechanism using solar energy and a wind turbine.

There are several proposed methods on LoRaWAN parameters performance in smart agriculture. The authors in [94] proposed a system-level analysis that confirms that the use of several gateways improves the network capacity and guarantees a higher success rate. The authors in [95] present an in-depth study of the performance of the LoRaWAN communication network to specify the adequate compromise between the number of nodes and the duration of the transmission interval. They propose a mathematical model that precisely predicts the successful packet delivery rate considering the number of nodes and the transmission interval duration. The proposed method predicts the behavior and performance of the LoRaWAN network and chooses the parameters during the design phase.

For effective and appropriate use of robots in industry, a robust and long-range wireless connection technology is required. The authors in [96] present a developed algorithm and approach to solving the problem of local navigation and localization of mobile robotic platforms based on LoRaWAN.

Providing maritime communications for ship/boat monitoring, maintenance of navigation marks, goods tracking, data extraction from sensing platforms, unmanned vehicle management, safety operations, voice communications, etc. is essential but it is a challenges task because of the lack of proper communication infrastructure and harsh environment of the harbor. In [97], authors present the different parts and typical operations in a harbor. Then, they investigate the LoRaWAN technology to locate and track assets in harbors. Their result shows that setting up a LoRaWAN infrastructure in a harbor environment for tracking purposes and geolocation is applicable and they mentioned that the user must derive the appropriate LoRaWAN parameters (like SF value, number of nodes, update rate, and coverage area) according to application and regulations. The authors in [98] present a real deployment to monitor Optimist Class sailing races for boat tracking and monitoring based on LoRaWAN. The proposed methods permit tracking not just each boat position but also key environmental and boat parameters. In the evaluation section, the authors discuss the good levels of coverage and link reliability with limited power consumption.

Implemented IoT systems in industries are faced with harsh and dynamic weather conditions. Their environments lead to increase location estimation error. Therefore, an acceptable localization method must properly identify environment error sources to improve the accuracy.

## 4.2. Smart Healthcare

Healthcare infrastructure improvement is one of the effective factors in increasing the quality of life. In the illness or aging period, continuous and long-term health monitoring systems are essential. A health monitoring system leads to better identification and treatment of the disease. Health monitoring and illness management are tasks that can handle by proper and expedients personals. In addition, it needs to continue presence at the hospital that makes it expensive. With the advent and development of IoT, smart healthcare has become a potential solution. The authors in [99] mentioned automatic identification and tracking of people and biomedical devices in hospitals, correct drug-patient associations, real-time monitoring of patient's physiological parameters for early detection of clinical deterioration as a few of the possible examples of smart healthcare. To improve the smart healthcare performance, several works suggested the use of wireless, small, portable, and externally wearable sensor nodes for health monitoring to remove specified location restrictions. The existing method proposed short-range communication (Bluetooth) for transferring between sensor data and a smartphone with a long-range communica-

tion (LTE) to transfer the processed information from the smartphone to the healthcare provider [100]. The most important drawback of these systems is power consumption. The authors in [101,102] presented a smart healthcare system for monitoring of human body signals that are performed by biomedical sensors, MySignals, and the LoRaWAN wireless network system. They evaluate the power consumption and battery life of the proposed method with Wi-Fi and BLE systems.

Smart healthcare systems have become more important for elderly people who live at home alone. They can fall in an emergency and lose consciousness. The authors in [103] mentioned that elderly people should be monitored and if specific events occur, emergency services should be automatically notified about event description and their location information. Therefore, they discussed a potential use of the LoRaWAN technology in healthcare, focusing on battery-powered end nodes with low power consumption.

Localization Methods in Smart Healthcare

LoRaWAN-based localization methods are used in different branches of healthcare systems (elderly care, search and rescue in mountain and sea, natural disaster, etc.). In this section, proposed methods in each branch are reviewed.

Recent advances in medical technology have led to population increment of elderly people. Elderly people face various diseases that need permanent healthcare systems and an accurate localization system to locate elderly people at the accident time. In [104], a comprehensive study is provided of the research and skills related to Active and Assisted Living (AAL) systems. The authors in [105] evaluate the widespread LoRaWAN communication infrastructure to alert a remotely connected healthcare server about elderly people accidents able to monitor the position and motion type of the elderly. They propose to enhance the capability of a LoRaWAN device for implementing an elderly friendly system platform using a wearable device equipped with a motion sensor.

Healthcare systems for workers in risky outdoor settings attract the attention of researchers. The authors in [106] mentioned that even a small accident may be dangerous if it happened in a remote place or during extreme weather conditions that make it difficult for the injured individual to seek help. Therefore, they propose a wearable LoRaWAN-based system for remote safety monitoring in remote areas with no network coverage to detect possible heart problems and/or a "man-down" situation. It then transmits an emergency alert containing information about the state of the concerned individual and its location via LoRaWAN to the surrounding recipients.

In [107], the authors designed and implemented a sea rescue system by using LoRaWAN in such an application and shows how it can be used to transfer data from a device attached to a Personal Flotation Devices (PFD), such as life jackets, to a receiver placed in a sea rescue helicopter. User in the proposed method equipped with a pulse sensor, to determine the health status of the user.

During a natural disaster, setting up a wireless communication infrastructure for enabling emergency message exchange among individuals and rescue teams for people localization and situations determination is necessary. Several works proposed a LoRaWAN base communications system for victim message exchange and localization [108–110]. Author in [111] proposed a phone-based Emergency Communication Systems (ECS) which allows multihop dissemination of emergency messages over LoRaWAN links. They investigated how to implement ECS able to leverage the pervasiveness of smartphones, and on the excellent propagation characteristics of the LoRaWAN technology to support data exchange even in highly critical scenarios. In [112], they extend their work and proposed a novel multihop dissemination algorithm that maximizes the probability to deliver an emergency request to the destination.

The authors in [113,114] proposed a LoRaWAN-based system for mountain search and rescue (SaR) operations. In the proposed method, the localization of the persons is obtained through an algorithm based on path-loss measurements. For doing this, they characterized the LoRaWAN Path Loss in three relevant mountain scenarios.

In smart healthcare systems, the accuracy and battery life of the proposed method is so important. There are different proposed methods on network architecture based on multi-modal and multihop nodes for collecting and transmitting environment sensor data with more effective latency and battery life. In all published studies until now on multihop network topologies, radio duty cycling is not considered. We think that a content-aware system for radio duty cycle determination is a requirement.

*4.3. Smart Transportation (Intelligent Transportation)*

With the growth of cities, the scale of the transportation system of cities has become large that leads to uncontrolled growth in traffic volume. The caused problems by increased traffic can be divided into three categories: traffic congestion (delay and fuel prices increment), safety (accidents and emergencies), and pollution (air pollution increment) [115]. IoT development has become a new opportunity for the elimination of transportation problems by introducing smart transportation [116]. Smart transportation develops a wide variety of sensors to gather information about the roads, vehicles, and drivers. Using these data, transportation servers locate vehicles and determine the best route for problem reduction.

For efficient development of smart transportation, authors in [117] proposed six different categories of sensors (safety, diagnostic, traffic, assistance, environment, and user) that can divide into two parts of in-vehicle sensors and road sensors. Effective use of sensors greatly depends on adopted wireless communication technology. The authors in [118] proposed a vehicle-to-everything (V2X) communication architecture with LoRaWAN wireless technology. The authors mentioned that LoRaWAN plays an important role in developing and solving fundamental challenges such as reliability, data handover, time criticality, modularity, and energy efficiency with vehicles on the move for V2X.

Performance of LoRaWAN schemes with different parameter configurations for V2X is evaluated and compared in [119]. The authors mentioned that higher BW and lower SF parameter configurations should be selected to resist the fast fading caused by the Doppler effect.

Localization Methods in Smart Transportation

One of the most popular categories of LoRaWAN base localization is vehicle monitoring, locating, and tracking. In this category, embedded sensors communication in vehicles is used for localization. This is useful for problems avoidance such as congestion. Commonly used methods for localization have limitations such as coverage, availability, and operational cost.

The authors in [120] mentioned that the availability of communication networks is an important part of the vehicle positioning system and the communication will be lost for the area which is not covered by the cellular network. To overcome this problem, they proposed the Integration of LoRa-Cellular (ILC) using hybrid communications to trace a vehicle position. The decision on which network is used for communication is decided based on the RSSI level and energy consumption.

The authors in [121] developed a transit vehicle tracking service prototype based on an Intelligent Transportation Systems (ITS) architecture to evaluates the operation of LoRaWAN technology in a transit vehicle tracking service in a medium-sized city. They investigate optimal LoRaWAN configuration parameters (SF, BW, and CR) for the service.

The authors in [122] proposed a real-time bus positioning system based on LoRaWAN technology which can help people to identify the dynamic location of buses and the expected arrival time. The proposed system is composed of terminal devices, data concentrators, cloud servers, and a user interface. The evaluation of the method show wide coverage, low loss rate, low power consumption, and low cost.

The authors in [123] propose an IoT-based bus location system using the LoRaWAN concept. In the proposed method, the location information of the bus is collected and distributed using the LoRaWAN network. In addition, they propose the use of smart bus stops using electronic paper, which has the characteristics of excellent outdoor visibility and low power consumption.

The authors in [124] proposed the development of an IoT-based public vehicle tracking system, using LoRaWAN and Intelligent Transportation Systems services. As a proof of concept, the authors designed and executed some experiments based on the proposed method. They identified in the experiments important technical aspects of LoRaWAN operation (LoS, SF, data rate) that must be considered.

### 4.4. Smart City

With the expansion of information and communication technologies, the smart city developed to improve the quality of life (QoL). The purpose of a smart city is to adopt modern technologies (Cloud/Edge computing, Cyber-Physical Systems (CPS), IoT, big data, security protocols, Information communication and technology, Artificial Intelligence (AI), Blockchain, and Geospatial technology) to convert every entity of a conventional city into an autonomous object performing its operation automatically without any substantial external help [125]. The authors in [126] mentioned four important characters of a smart city consist of sustainability, QoL, urbanization, and smartness. A smart city is composed of different components depending on the areas of interest. Smart industry, smart transportation, smart healthcare, and smart community. The first three mentioned components are considered and discussed separately in this paper due to their importance. In this context, the smart community covers smart city assets and services management.

Transmission layer of the smart city act as a backbone that provides communication between other layers by adopting communication technologies. A communication technology with a low cost of deployment, low power consumption, and long-range coverage makes the deployment of smart cities more efficient. The authors in [127] mentioned that commonly used cellular networks suffer from the high cost and poor battery life. In addition, local RF technologies, including Bluetooth and Wi-Fi, do not meet the range requirements to support smart cities applications. Therefore, the authors design and implement a wireless sensor network based on the LoRaWAN protocol for a smart city.

The authors in [128] optimize the transmission parameters of a LoRaWAN system in a smart city environment. Their approach significantly improves the success rate and enables more nodes to use lower spreading, which results in lower delay and power consumption.

Localization Methods in Smart City

LoRaWAN technology provides robust, non-expensive, and effective methods for smart city assets localization. On the other side, it provides a long-range position method for lost people. In this subsection, a review of the proposed method for lost people and assets positioning is provided.

The authors in [129] point out that for an advanced smart city, technology may remotely monitor, optimally control (like street lighting, the timing of traffic lights, access to and price of public parking, garbage collection) and track the location and movement of public assets (like garbage collection vehicles and roadway barriers). The authors address the problem of using IoT technology within a smart city to track the location of relatively small and inexpensive non-powered assets. In the evaluation, four objectives of Cost (asset tracker costing less than 5% of the resource with which it is partnered), Distance (2.5 km), Durability, and Sustainability (operation for 5+ years without manual intervention in most geographical areas) is addressed. Bike-sharing systems are another part of cities and bikes are assets and resources of cities. The authors in [130] mentioned that one of the problems that afflicts bike-sharing systems are the loss of bikes. They study in simulation the scalability limits of a typical LoRaWAN for bike-sharing systems and showed that the performance of LoRaWAN in crowded scenarios can be quite limited when using high SFs and that fading has a moderate impact on the cell capacity. Then, they design and realize a prototype of a LoRaWAN tracker module that can be embedded in a bike and test its use in a large area city. On the other hand, LoRaWAN is an effective technology for object tracking as well in a smart city. The authors in [131] aims to verify the possible use of the LoRaWAN for mobile applications with attention to timing performance, which can greatly affect the

overall service. In particular, an experimental setup mimicking real-world scenarios has been purposely implemented using the LoRaWAN infrastructure. In this way, the delay introduced by the communication infrastructure from the field up to the data sink at the application level has been evaluated. Ecotourism such as national forest parks is becoming increasingly popular. However, the accidents ratio as with getting lost and the occurrence of natural disasters is growing too. The large area of ecotourism makes LoRaWAN an acceptable communication technology for positioning. The authors in [132] propose a system for monitoring and assisting visitors of forest parks based on LoRaWAN for the development of a communication infrastructure capable of allowing communication. The proposed method is used to support an application for monitoring visitors' location.

### 4.5. Comparative Analysis and Discussion

A summary of the different requirements of IoT systems is presented in Table 3. Location estimation accuracy and latency in smart factories and healthcare systems have high priority. Smart factories and healthcare are critical and hard real-time systems. A location error or latency higher than an acceptable threshold can lead to safety harms while lower location estimation accuracy in smart transportation and smart cities is acceptable. Energy consumption in smart healthcare and the smart city because of buttery restriction is a limitation. Smart transportation and smart city have a dynamic with high mobility speed environment. Therefore, the proposed method for them should be scalable to handle their dynamic environment.

**Table 3.** Summary of different requirement of IoT systems ($\checkmark$ = necessary requirement, * = unnecessary requirement).

| Application Domain | Accuracy | Latency | Energy Consumption | Scalability | Security and Privacy |
|---|---|---|---|---|---|
| Smart Factory | $\checkmark$ | $\checkmark$ | * | * | $\checkmark$ |
| Smart Healthcare | $\checkmark$ | $\checkmark$ | $\checkmark$ | * | $\checkmark$ |
| Smart Transportation | * | * | * | $\checkmark$ | $\checkmark$ |
| Smart City | * | * | $\checkmark$ | $\checkmark$ | $\checkmark$ |

A summary of the different proposed methods for IoT-based systems localization is provided in Table 4. A fundamental challenge of Industry 4.0 is the presence of multipath effects. Different proposed methods in this category try to improve accuracy by adopting different technologies but there is a need to determine the environment error source and their effect on the signal. Accuracy and power consumption are the most effective factors for a smart healthcare system. Most of the proposed methods in this category try to improve location estimation accuracy using GPS technology but it leads to increase power consumption. Low mobility and environmental dynamics make it possible to determine and predict the effect of environment error sources on signals to improve accuracy and power consumption simultaneously. Due to the highly dynamic environment of smart transportation, system latency reduction, and scalability. By adopting different technologies, proposed methods try to provide context-aware solutions and improve these factors. Finally, as power consumption efficiency in a smart city is an important factor, multi-technology networks and resource-aware have emerged as an important technology to improve this factor [133–136].

**Table 4.** Summary of different proposed method for IoT systems localization.

| Reference | Application Domain | Purpose | IoT Localization Approach |
|---|---|---|---|
| Rohit et al. (2018) [90] | Industry 4.0 | Asset localization and tracking | GPS and LoRaWAN |
| Swain et al. (2021) [93] | Industry 4.0 | Agriculture sensor nodes localization | LoRaWAN and Wi-Fi |
| Miles et al. (2020) [95] | Industry 4.0 | Performance evaluation of the LoRaWAN | Number of Nodes and transmission interval |
| Iakovlev et al. (2020) [96] | Industry 4.0 | Mobile robotic localization | UAV aided with LoRaWAN |

**Table 4.** *Cont.*

| Reference | Application Domain | Purpose | IoT Localization Approach |
|---|---|---|---|
| Priyanta et al. (2019) [97] | Industry 4.0 | Harbors assets localization | Evaluate SF, number of nodes, etc. parameters |
| Sanchez et al. (2019) [98] | Industry 4.0 | Boat tracking and monitoring | RSSI with different LoRaWAN factors |
| Fernandes et al. (2020) [105] | Smart Healthcare | Elderly position monitoring | GNSS for outdoor and ToA for indoor |
| Tayeh et al. (2020) [106] | Smart Healthcare | Localization in risky outdoor | GPS with LoRaWAN |
| Dehda et al. (2019) [107] | Smart Healthcare | Sea rescue system | GPS with LoRaWAN |
| Sciullo et al. (2018) [111] | Smart Healthcare | Phone-based ECS | GPS with LoRaWAN |
| Bianco et al. (2020) [113] | Smart Healthcare | Mountain Search and Rescue | Pass loss model |
| Ilham et al. (2019) [120] | Smart Transportation | Hybrid communications | RSSI level and energy consumption |
| Jurado et al. (2020) [121] | Smart Transportation | Transit vehicle tracking | RSSI |
| Guan et al. (2018) [122] | Smart Transportation | Bus positioning system | GPS with LoRaWAN |
| Boshita et al. (2018) [123] | Smart Transportation | IoT-based bus location system | using the location and delay information |
| Salazar et al. (2020) [124] | Smart Transportation | Public vehicle tracking | GPS with LoRaWAN |
| Deese et al. (2020) [129] | Smart City | City assets track and localization | RSSI |
| Croce et al. (2019) [130] | Smart City | Bike localization system | RSSI |
| Carvalho et al. (2018) [131] | Smart City | Mobile applications localization | Network transfer protocol |
| Ferreira et al. (2020) [132] | Smart City | Ecotourism visitor assistance | GPS and RSSI |

## 5. Conclusions

In the near future, IoT-based systems will have an important role in several application domains and common IoT communication technologies will be broadly disseminated. For example, LPWANs technologies may be seen as a key technology enabler in several IoT domains due to their low implantation cost, and high compatibility with resource-constrained devices. Moreover, they provide long-range communications with low power consumption. Presently, three major LPWAN communication technologies—LoRaWAN, Sigfox, and NB-IoT—are being widely used in several IoT application domains, which have been covered in detail in this article, namely: smart cities, smart transportation, smart healthcare, and smart manufacturing. Furthermore, each application domain has its own different singularities and requirements, being the provision of location information an important feature in asset tracking and management. Using LPWAN technologies, it is possible to perform communications and localization simultaneously, reducing not only the implantation cost but also the IoT device energy consumption. More specifically, the LoRaWAN technology presents several important strengths that make it more appropriate for communication and localization in IoT-based ecosystems. Therefore, this article focuses the exploration of low-cost GNSS-free outdoor localization techniques, with a focus on the LoRaWAN technology. Following, the four main open issues of IoT-based systems that affect the provision of location information are introduced and discussed:

- Energy Consumption: it is an important limitation in IoT-based systems. The main portion of energy consumption is related to connectivity, typically during the data transmission periods. To decrease data transmission, resource-aware and context-aware solutions are becoming an attractive subject for researchers [137].
- Security and Privacy: LPWAN technologies contain important security and safety vulnerabilities These vulnerabilities are exploited by malicious entities and lead to great damages. This becomes more important in some IoT categories (Industry 4.0, smart city) and it is a drawback. Therefore, providing a reliable security mechanism based on their limitations is a challenging and open issue task [15,138].

- Accuracy: there are different methods for accuracy improvement in positioning methods, but outdoor localization for resource-constrained IoT devices is still an open issue. Several IoT-based systems deployed in different environments have different error sources that should be considered to improve localization accuracy [39].
- Scalability: by increasing device density, for example, the scalability of a LoRaWAN network becomes a challenging issue. This is more important in a smart city application domain, which typically presents a dynamic and large number of end devices. Moreover, the scalability of a LoRaWAN network can be affected by different factors including co-SF interference, inter-SF interference, and the class selected during the design and implementation of the end nodes [139].

Based on the identified open issues, future work directions should focus on the improvement of the limitations of LPWAN-based localization methods without forgetting to address each application domain specificities.

**Author Contributions:** Conceptualization, A.K. and S.I.L.; Methodology, A.M., A.K. and S.I.L.; Investigation, A.M., A.K., S.I.L.; Writing—Original Draft Preparation, A.M., A.K. and S.I.L.; Writing—Review and Editing, H.R. and S.P.; Supervision, A.K. and S.I.L.; Project Administration, A.K. and S.I.L. All authors have read and agreed to the published version of the manuscript.

**Funding:** This work has been partially supported by the TECH (Norte-01-0145-FEDER-000043) project, and the CoViS (POCI-01-02B7-FEDER-070090) project, under the PORTUGAL 2020 Partnership Agreement, funded through the European Regional Development Fund (ERDF).

**Institutional Review Board Statement:** Not applicable.

**Informed Consent Statement:** Not applicable.

**Data Availability Statement:** Not applicable.

**Conflicts of Interest:** The authors declare no conflict of interest.

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
