# Peer review of "GNSS-Free Outdoor Localization Techniques for Resource-Constrained IoT Architectures: A Literature Review"

_applsci, doi:10.3390/app112210793_

Round 1
Reviewer 1 Report
The authors perform a survey of localization solutions not based on GPS on IoT networks.
Comments:
I should avoid abbreviations in the title!! GNSS.
In page 2, line 83 the authors classify as LPWAN Sigfox, LoraWAN and NB-IoT, but there are other technologlies classified as LPWAN, such as LTE-M.
Figure 1 is placed far from it was referenced. Also, this figure requires further explanation.
In Table 1 (authors call it “Tabela”) the authors summarize LPWAN technologies. However, LTE-M should be included. In addition, RF spec (for instance South America and other continents) should be considered for the different technologies, as well as features as packet sizes and energy consumption features. Also, the caption of this table should be improved.
Figure 2 is placed far from it was referenced.
Figure 4, we cannot see the detail. The letter size should be at least the same size as the caption.
In section 4, only LoraWAN is considered, however, there were introduced other technologies as LPWAN.
In Table 2 (authors call it “Tabela”) the caption should be more specific and further detail. Also, there are abbreviation that should be used once defined such as DNN.
In Table 3 (authors call it “Tabela”), this information is only based on LoraWAN. If no other technologies are going to be included (we would recommend to include other classified as LPWAN), this should appear in the caption.
In section 5, conclusion, it should state that this is focused on LoraWAN only, as mentioned before.
Other references to be included to improve the survey:
-regarding Fog computing architecture, a reference regarding orchestration with containers would help the reading, such as:
Performance comparison of container orchestration platforms with low cost devices in the fog, assisting Internet of Things applications, Journal of Network and Computer Applications,
https://doi.org/10.1016/j.jnca.2020.102788
-in order to motivate localization in the same context, without GPS, the next reference is relevant for section 3.3. Localization Accuracy Improvement Methods
Indoor localization using time difference of arrival with UWB signals and unsynchronized devices,
Ad Hoc Networks, https://doi.org/10.1016/j.adhoc.2019.102067
Minor issues:
-Define IIoT (keyword) and not defined in the paper
-pag 6, line 281 typo: “.. it Provides ..”-> it provides
-pag 7, line 320 typo: “communication. it leads” -> “communication. It leads”
-Tabela -> Table
Author Response
We thank you for your consideration and the time you spent on our paper. We have revised the manuscript according to your comments and suggestions. Attached file is the point-by-point responses to your comments. We hope our revision may improve the paper.

Reviewer 2 Report
The authors survey the GNSS-free outdoor localization techniques, in particular, LPWAN. The article is interesting, however, there are some points of the documents that bring me confusion. Section 2.4 for instance is a general overview of Localization Systems (LS) architectures. There is no analysis of architectures for LS. How these kinds of architectures are correlated with the systems? Also, section 2.4.1 describes a cloud architecture in general, without entering in detail with the localization systems requirements. [1]. How these two architectures (cloud and fog) affect the localization systems features?
Section 2.2 must consider the requirements of LS and not IoT general IoT systems. Moreover, an appropriate privacy-by-design architecture must be developed [2] and should be mentioned in the “security & privacy” requirement. About “energy consumption”, IMHO the most important aspect is to exploit the wireless interface already presented in several mobile devices and used for communication purposes, also to localize the devices themself.
Section 3.1. There are different kinds of propagation models. Two-ray [3], Nakagami-m, gamma distrubutions etc... [4]. Please mention some works about these models, and describe why you believe that is more important the log-normal model. Moreover, equation (1) has not the gaussian (normal) distribution component.
Minor comments:
- About smart health discussion (sect 4.2). One of the first works in this field is [5]. It is more related to AAL. But can be interesting for this paper.
- Pg 10 “achieved performance and cost [30]”. Also maintenance cost, i.e. the time spent to maintain the LS is important.
- Pg 7. “communication. it leads to a”. Capital letter
- Pg 6. “license-free spectrum and it Provides”. Capital letter
- Pg 6. “. By Raising TX-PoW, the probability of signal resistance versus channel fading increases.”. What does it means this sentence? Why raising the transmission power the channel fading increases? Please reference.
- pg. 5. “As a result, By increasing the BW, chip duration would increase accordingly”. For me, it is exactly the contrary. If BW increases, and BW is equal to the inverse of Tc, then Tc decreases.
[1] “Discovering location based services: A unified approach for heterogeneous indoor localization systems”
[2] “A privacy-by-design architecture for indoor localization systems”
[3] “Measurement-based frame error model for simulating outdoor Wi-Fi networks “
[4] “Channel models for terrestrial wireless communications: a survey”
[5] “Evaluating ambient assisted living solutions: The localization competition”
Author Response

(The authors gave the same response as above.)

Round 2
Reviewer 2 Report
The authors respond to all my comments.
Minor issue:
The authors wrote "chip duration and chirp duration would change accordingly", probably is "symbol duration and chirp duration would change accordingly"
Author Response
Answer 1): Dear reviewer, thank you for your comment. The mentioned sentence has been revised to “symbol duration and chirp duration would change accordingly”.